# In-Context Learning for Esoteric Programming Languages: Evaluating and Enhancing LLM Reasoning Without Fine-Tuning

## Abstract

Large Language Models (LLMs) have revolutionized mainstream software development, yet their ability to generalize to esoteric languages—who may have small or no representation in the training corpus—remains poor. Programming in esoteric languages tests a model's capacity to infer novel grammar and leverage nontrivial reasoning capabilities in utilizing the documentation. To quantify these effects, we evaluate both open and closed-source LLMs on code generation and language identification tasks across four esoteric languages—Minipy, Pyth, Rhokell, and 0815—and compare traditional prompt-based methods to agentic coding IDEs. Our findings reveal that LLMs can now generate some correct code in these languages when provided with documentation and sparse examples; however, performance remains far below that of similar models in common programming languages. Furthermore, we introduce a novel in-context augmentation strategy in which LLMs first generate solutions, which are then verified and re-inserted as examples into subsequent prompts. Our results indicate that strategically embedding just a few analogous problems can yield large accuracy improvements without any model retraining. Our findings show that this "self-scaffolding" approach can boost performance on coding benchmarks: inserting Deepseek's verified EsoEval solutions raised EsoEval accuracy on Pyth from 16.67% to 30.82 %, while HumanEval accuracy on Minipy jumped from 51% to 65%. We offer this as a flexible alternative to costly fine-tuning, paving the way for rapid adaptation of LLMs to highly specialized, emerging, or other low data domains.

## 1 Introduction

Large Language Models (LLMs) pretrained on massive amounts of text and code data have demonstrated promising performance across various code generation tasks. As these models become increasingly prevalent, a key application area is the generation of code in specialized domains. New languages are constantly being developed to better address things like performance, security, or ease of writing specific types of programs. Translation and maintenance of legacy code can also drive the need for expertise in more obscure programming languages. Given the high cost of fine-tuning LLMs, in-context learning—leveraging instructions and examples provided within prompts—has emerged as the preferred method for adapting these models to tasks and domains that were not encountered during training.

Previous studies have employed in-context demonstrations to prompt LLMs to generate code that interfaces with external, task-specific library functions Gupta and Kembhavi (2023b); Patel et al. (2024). Additionally, Patel et al. observed that LLMs exhibit a strong capability to understand and utilize novel code libraries based solely on in-context information. Remarkably, their work showed that LLMs could learn an entirely unfamiliar programming language, Isabelle, even though there is minimal available data on it online. This indicates that LLMs are capable of combining non-trivial reasoning skills with the syntax of a new language learned entirely through contextual examples.

Building on the preliminary observation that LLMs can learn new programming languages from scratch using only in-context demonstrations, we aim to explore several key questions about this phenomenon. Specifically, we investigate: which esoteric languages can LLMs effectively handle?

How far can these languages deviate from conventional programming paradigms while still being learned effectively? Do smaller, open-source models exhibit similar capabilities? In this work, we outline our evaluation framework and present our findings addressing these questions.

Beyond assessing LLMs' ability to learn esoteric languages (esolangs), our investigation provides broader insights into their generalization capabilities in low-resource code generation settings. Unlike mainstream programming languages, which benefit from extensive online documentation and training data, esolangs present a challenging test bed where models have had far less data to learn from but can rely on full documentation provided at run-time. Understanding how LLMs navigate these constraints can inform strategies for improving in-context learning in practical scenarios; by probing the limits of LLMs in such unconventional domains, our study sheds light on both their strengths and potential failure modes, contributing to a deeper understanding of their inner workings and future improvements in code generation models.

## 2    RELATED WORK

Recent studies have made significant progress in enabling large language models (LLMs) to generate code from in-context prompts, even when using unfamiliar libraries or syntaxes.

One prominent direction is retrieval-augmented code generation, where external documentation or code is provided as part of the prompt. For example, *DocPrompting* by Zhou et al. (2023) retrieves relevant API documentation and adds it to the model's prompt, helping LLMs adapt to unseen libraries without retraining. Similarly, Hsieh et al. (2023) show that supplying tool documentation can enable zero-shot tool use, matching or exceeding few-shot performance without requiring explicit demonstrations.

Another important line of work focuses on optimizing which examples to include in few-shot prompts. Li et al. (2023) propose *Large Language Model-Aware In-Context Learning*, a technique that selects in-context examples based on how much they boost the model's likelihood of solving the task. This leads to substantial gains over traditional retrieval strategies. Complementary to this, Li et al. (2024) introduce *AceCoder*, a staged prompting approach where LLMs are asked to first generate a high-level problem analysis before writing code, further improving code generation accuracy across multiple benchmarks.

In addition to static prompts, dynamic retrieval strategies have been explored. Su et al. (2024) propose *EVOR*, an evolving retrieval framework where the model iteratively refines its retrievals based on generated partial code and execution feedback. EVOR demonstrates significant gains on tasks involving frequently updated libraries and obscure programming languages compared to traditional static retrieval methods.

The question of whether LLMs can learn novel libraries and programming languages purely from in-context information has been explicitly studied by Patel et al. (2024). Their evaluation shows that LLMs can effectively understand and use previously unseen APIs when provided with either usage examples or plain text descriptions. However, they focus on domain specific tasks testing vision recognition libraries and the language Isabella for automated theorem proving. We are interested in broad programming abilities and examine multiple programming languages with differing properties. Similarly, Gupta and Kembhavi (2023a) demonstrate that LLMs can generate compositional programs by observing a new vision-language API without any task-specific fine-tuning.

(Athiwaratkun et al., 2023) introduce MBXP and Multilingual HumanEval, execution-based benchmarks that evaluate code generation in many mainstream programming languages by converting existing Python datasets such as MBPP and HumanEval into target languages. Their framework shows that large multilingual models can generalize across languages and benefit from cross-lingual training, and that new languages can often be handled via few-shot prompting alone.

Finally, Mora et al. (2024) explore a different setting: enabling LLMs to handle very low-resource and formal languages through synthetic intermediate representations. Their method, *SPEAC*, improves LLM performance by constraining generation to a repairable pseudo-language that can later be compiled into the target formalism.

Whereas previous studies focus on a single API or language, we evaluate LLMs across a diverse range of esoteric languages (Minipy, Pyth, Rhokell, 0815) that vary in syntax and online footprint, and under

two benchmarks (HumanEval, EsoEval) of differing complexity. We introduce a self-scaffolding procedure: model-generated solutions are automatically verified and then re-inserted as in-context examples, to boost performance without retraining, providing a lightweight adaptation method that complements these prior techniques.

# 3 METHODOLOGY

## 3.1 MODEL FAMILIARITY

| | Minipy | 0815 | Rhokell | Pyth |
|---|---|---|---|---|
| gpt-4o-mini | / | ✗ | ✗ | ✓ |
| gpt-4o | / | ✗ | ✗ | ✓ |
| Deepseek-V3 | ✓ | ✗ | ✓ | ✓ |
| LLAMA-3.3-70B | ✓ | ✗ | ✗ | ✓ |
| Agentic 4o | ✗ | ✗ | ✗ | ✓ |
| Agentic Claude | ✗ | ✗ | ✗ | ✓ |

Figure 1: Level of familiarity each tested model has with our Esolang dataset.
/ = Attempted definition of the language, but vague—could likely be guessed just from context of the name of esolang
✗ = No familiarity with the language
✓ = Clear understanding of the language

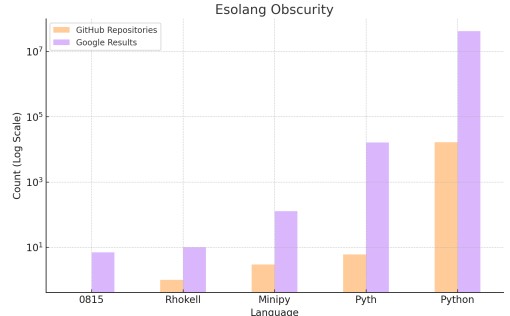

Figure 2: Obscurity of our esoteric programming languages, along with Python as a reference. We see the selected languages being orders of magnitude less common than Python.

We conducted an assessment of our model's familiarity with our chosen esolangs by asking the models to describe the language and to identify code examples. We evaluated the model's descriptions of the programming languages by hand, as summarized in Figure 1. For the code identification task none of the models were able to correctly identify the programming language from the code examples.

## 3.2 MEASURING ESOLANG OBSCURITY

To better understand the challenge posed by evaluating LLMs on esoteric programming languages, we first quantify how obscure these languages are. Esoteric languages vary significantly in online presence, documentation, and community adoption. To assess their obscurity, we gathered data on two key indicators: (1) the number of search engine results containing references to each language and (2) the number of publicly available GitHub repositories referencing this language.

**Search Engine Presence** To estimate the prevalence of each esolang in search engine results, we queried the phrase "X" "programming language" (where X is the language name). Since raw search result counts can be unreliable due to noise, we refined our approach by examining a random sample of approximately 50 pages from the top 100 results. Each sampled page manually classified to determine whether it was genuinely about the given programming language or an unrelated topic. We then used the proportion of relevant results to extrapolate an adjusted estimate of the total number of valid search hits. This likely gives an overestimate as earlier search hits are more likely to be relevant.

**GitHub Presence** To assess the extent to which each esolang is actively used in coding projects, we performed a GitHub search using the same query format ("X" "programming language") to identify repositories mentioning the language. While GitHub does not provide precise counts of repositories containing code in pit esolangs, we believe this still provides a useful estimate of community engagement and adoption.

**Obscurity Measurements** Figure 2 show the stark contrast between mainstream programming languages and esoteric ones in terms of online presence. As a reference point, Python has approximately 42 million adjusted Google search results and 16,500 GitHub repositories, whereas Pyth, our best known language, has only 16,200 Google results and 6 repositories (1000-10000x fewer examples). This disparity underscores how infrequently LLMs are likely to encounter these esolangs in their training data.

## 3.3 BENCHMARKS

We form a benchmark that is language agnostic, Esoeval. While numerous benchmarks exist to evaluate the general code generation abilities of language models, we do not use popular benchmarks like SWE-bench Verified, which is currently favored but limited to Python, or MBPP, which also targets only Python and contains mostly simple synthesis tasks. Both of these benchmarks involve fixing or completing code already written in Python or explicitly asks for Python code. These benchmarks are not language-agnostic, making them less suitable for evaluating model performance across diverse or obscure programming languages—our primary focus. Thus, we focus on HumanEval and our novel EsoEval benchmark.

**HumanEval** This hand-written evaluation set consists of 164 programming problems, each including a function signature, docstring, function body, and unit tests (an average of 7.7 per problem).

**EsoEval** The HumanEval dataset was quite challenging for LLMs to code in, so we generated an additional baseline for comparison. We present EsoEval—a simplified set of 100 problems. EsoEval includes tasks ranging from basic output statements (e.g., printing "Hello world") to more complex logic problems (e.g., computing factorials, evaluating prime numbers, and performing string manipulations). Despite the complexity variations, these tasks remain relatively simple. To establish a baseline, we evaluated EsoEval in Python using OpenAI's gpt-4o-mini, which achieved a 100% accuracy rate, confirming that these tasks are suitable for standard benchmarking.

## 3.4 MODELS

We experimented with a range of models, including GPT-4o-mini, GPT-4o OpenAI (2024),LLAMA-3.3-70B-Instruct-Turbo Grattafiori et al. (2024), Deepseek V3 Liu et al. (2024), and agentic IDEs, e.g. Codeium's Windsurf Codeium (2025). We evaluated a range of open and closed source models.

## 3.5 PROMPTING

> **Standardized Prompt Template**
>
> Write a function in *[esoteric language]*, an esoteric programming language. The function should perform the following: *[prompt]*.
> The documentation for [esoteric language] is provided here: [documentation].
> . . .
> **In-Context Examples.**

## 3.6 DOCUMENTATION/IN-CONTEXT EXAMPLES

Online documentation was assessed by hand and then reformatted if needed. In addition, we provided between five and thirteen in-context examples per esoteric language—drawn from simple, common tasks (e.g. factorial, Fibonacci, parity checks) and sourced from public GitHub repositories under verified fair-use.

## 3.7 EVALUATION

For the agentic IDE evaluation, ( i.e. Codeium's Windsurf Codeium (2025)), the setup only differs in the code generation step where programs were instead generated sequentially in a separate context to prevent cross-reference.

---

**Evaluation Workflow for Standard LLMs:**

1. **Prompt Augmentation.** The standardized prompt, augmented with the relevant documentation, is sent to the chosen code-generation API or language model to produce candidate code in the specified esoteric language.

2. **Code Extraction.** The response is parsed to extract the candidate esoteric code.

3. **Execution.** The extracted code is saved to a temporary file and executed using the esoteric language's interpreter via a subprocess call.

4. **Testing.** Input arguments and expected outputs are derived from the HumanEval test cases. The candidate code is executed with the provided inputs, and its output is compared against the expected output, allowing for minor formatting differences.

This methodology provides a robust framework for evaluating the ability of various models to generate and execute code in esoteric programming languages.

---

## 4 RESULTS

A series of experiments were conducted to investigate the capability of several LLMs to generate code in esoteric programming languages. Four primary esolangs were selected for evaluation—Minipy, 0815, Pyth, Rhokell each tested on two different benchmarks: the standard HumanEval dataset and a newly created simplified benchmark, EsoEval. Figure 3a provides a visual overview of the accuracy rates for each language–model pairing on EsoEval. Figure 3b provides a visual overview of the accuracy rates for each language–model pairing on Humaneval.

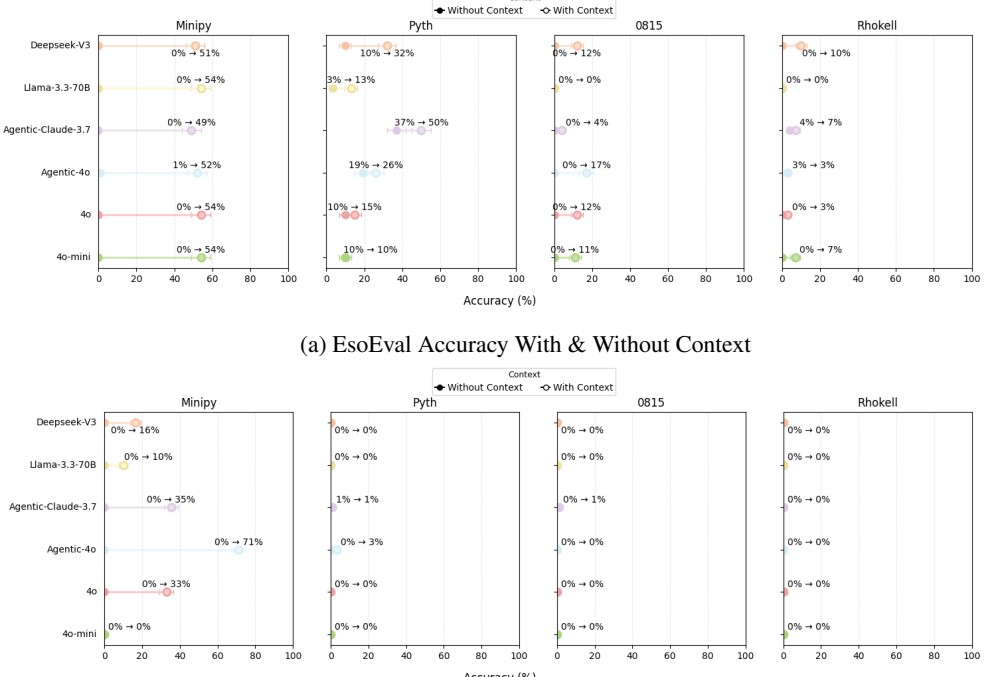

(a) EsoEval Accuracy With & Without Context

(b) HumanEval Accuracy With & Without Context

Figure 3: Accuracy of six language models on EsoEval and HumanEval across four esoteric languages, comparing performance with and without contextual examples/documentation. For Minipy we additionally required the use of some Minipy feature to be counted as correct.

We originally thought that a model's ability to produce code that compiles an unfamiliar esoteric language would serve as a rough proxy for its underlying "grasp" of that language's syntax and seman-

tics, and that higher compilability would therefore translate into higher accuracy. The scatter-and-fit lines above in Figure 4 , however, show almost little relationship: slopes are near zero in three of the four languages, and even in Pyth (where the slope is greatest) a rise in compilability from 20% to 80% yields only a 15-point boost in correctness. Models will occasionally learn just enough grammar to compile correctly, matching parentheses, using valid tokens and whatnot, yet still produce algorithms that don't solve the target problem.

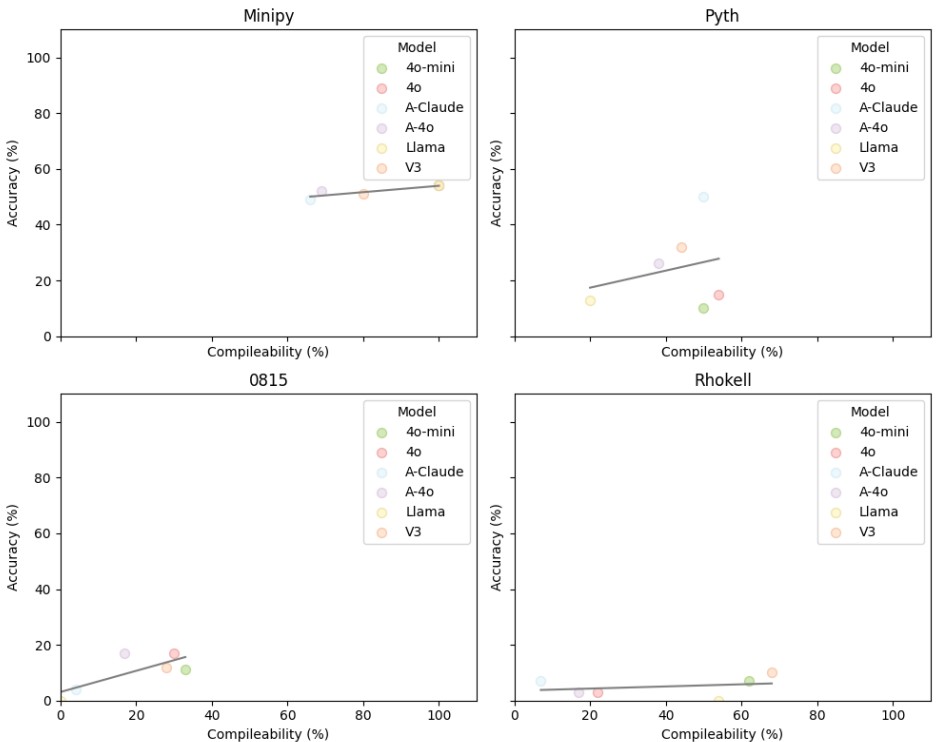

Figure 4: Accuracy vs. compilability on EsoEval. For Minipy we additionally required the use of some Minipy feature to be counted as correct.

This pronounced disconnect is striking when compared to mainstream benchmarks,take Python, for example, where a model's ability to compile is almost always a reliable indicator of its solution's correctness. In our own experiments on standard Python benchmarks, we found that nearly all generated solutions that compiled also passed all provided unit tests for EsoEval, making compilation success a robust proxy for functional correctness. In addition, most language models tend to perform similarly when tested on Python, demonstrating predictable trends in compilation and execution accuracy. This is likely due to well-documented syntax, extensive training data, and consistent execution environment. In contrast, our results on esoteric languages reveal no clear "winner"—each model's strengths vary sharply from one language to another.

### 4.1  IN CONTEXT AUGMENTED LEARNING

Our findings show that augmenting model prompts with in-context examples generated by the LLMs themselves can improve subsequent performance on difficult code-generation benchmarks, see Figure 5. By inserting correct EsoEval solutions into the context for EsoEval and HumanEval tasks, we observed DeepSeek's EsoEval accuracy on Pyth rise from 32% to 41 % and HumanEval accuracy on Minipy jump from 16.46 % to 30.82 %. None of the examples added to the HumanEval prompts overlapped with the EsoEval generated examples, demonstrating that our gains stem from the contextual scaffolding provided by similar, but not identical, problems. Similarily, we observed gains in GPT 4o with a noteable jump from 54% to 63% in Minipy on EsoEval and a jump from 32.92% to 39.02% in HumanEval. At each step, any new problems the model solved were added to

the next prompt, so it had more correct examples each time. Subsequent rounds of example insertion produced diminishing returns, this plateau may suggest that a relatively small number of well-chosen examples suffices to saturate the model's context-driven learning capacity.For example, when tackling complex or specialized problems, strategically embedding a few similar examples within the context window can potentially lead to enhanced accuracy without the need for extensive retraining. This approach offers a flexible, resource-efficient alternative to traditional fine-tuning, making it a valuable tool for adapting models to highly specialized tasks. This strategy offers a lightweight, adaptable pathway for extending large language models to highly specialized or emerging domains without costly retraining—a promising direction for resource-efficient model adaptation.

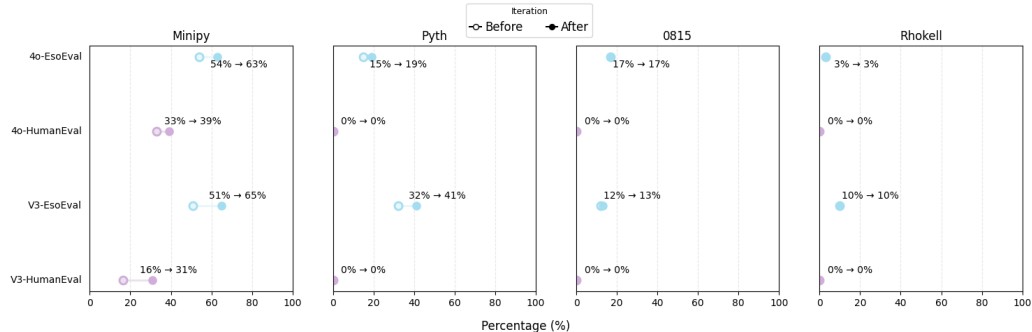

Figure 5: Accuracy after iteratively adding correct model generated examples to the context. This was repeated until there was no additional benchmark improvement. For Minipy we additionally required the use of some Minipy feature to be counted as correct.

## 4.2 LANGUAGE SPECIFIC OBSERVATIONS

Minipy occupies a unique position among our esoteric languages: although it extends Python with concise shorthand constructs, most HumanEval tasks do not require those extensions, so a model can "cheat" by emitting plain Python and still pass the tests. To prevent this shortcut, we enforced a non-Python compilability requirement in our EsoEval metric: any submission that successfully ran under a standard Python interpreter were excluded, regardless of functional accuracy.

When evaluated on HumanEval, GPT-4o-mini invariably fell back on plain Python, yielding 0% of solutions that failed to compile under a standard Python interpreter. Llama-3.3-70B-Instruct-Turbo exhibited the same tendency, with only 10 % (HumanEval) and 7.7% (HumanEval subset) of its outputs producing non-compilable code. By contrast, on the simpler EsoEval benchmark—where true Minipy syntax is required—both models showed dramatic gains in non-Python compilability accuracy: GPT-4o-mini reached 54% non-compilable submissions, Llama-3.3-70B-Instruct-Turbo 54%, and DeepSeek V3 51%.

These results suggest that, when confronted with complex tasks, models prefer the safety of familiar Python constructs rather than leverage Minipy's shorthand features. However, on more straightforward problems, they are capable of nontrivially applying the documented Minipy extensions. By measuring non-compilability in Python, we ensure that high EsoEval accuracy truly reflects understanding of Minipy's specialized syntax rather than a fallback to Python.

Across the other three esoteric languages (0815, Pyth, and Rhokell) and two evaluation frameworks (HumanEval, 10 tasks; EsoEval, 30 tasks), we observed that the degree of syntactic divergence is correlated with LLM performance. For example, the hexadecimal-only, comment-filtering 0815 language, GPT-4o-mini scored 0% on HumanEval but 11% on the simpler EsoEval benchmark, whereas LLAMA-3.3-70B achieved 0% and Deepseek V3 12% on EsoEval. In Pyth—a Python-inspired golfing language—GPT-4o-mini again scored 0% on HumanEval but attained 10% on EsoEval, with LLAMA-3.3-70B and Deepseek V3 reaching 13% and 32%, respectively. Finally, for Rhokell, which fuses $\rho$ calculus with Haskell-style syntax, GPT-4o-mini produced 0% accuracy on HumanEval but 3% on EsoEval, while LLAMA-3.3-70B remained at 0% and Deepseek V3 achieved 10%. These results suggest that moderate syntactic departures—such as Pyth's concise,

Python-derived abbreviations—permit some transfer of existing knowledge, but more unusual syntax like those of 0815 and Rhokell inhibit code generation.

### 4.3 WITHOUT CONTEXT:

We evaluated the EsoEval problems across four languages: Pyth, 0815, and Minipy—without providing any accompanying documentation or examples. By mandating non-Python compilability we show the context-dependence of their performance for most models. However, the relatively high accuracy observed for Pyth is concerning and may be attributed to the language's lower level of esotericism. When a model, such as gpt-4o-mini, achieves the same accuracy with and without context for Pyth, it is likely due to exposure to similar samples in its training data, thereby diminishing the extent of true in-context learning.

### 4.4 RELATIONSHIP BETWEEN OBSCURITY AND PERFORMANCE

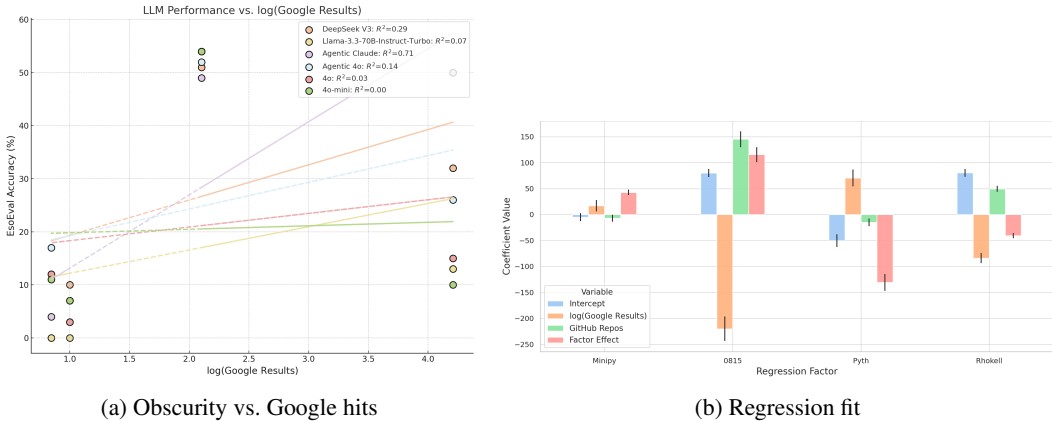

(a) Obscurity vs. Google hits          (b) Regression fit

Figure 6: Correlation Between Obscurity and Performance

The experiments indicate that closer syntactic parallels to Python lead to higher accuracy rates, as evidenced by Minipy and, to a lesser extent, Pyth. Non-standard languages such as 0815, which substantially differs from standard languages, elicited near-zero accuracy on more complex tasks. Incorporating relevant documentation within the prompt proved to be an effective strategy for improving the models' ability to generate valid esolang code. It appears that obscurity—measured here as the log of Google result counts—has little bearing on a model's ability to generate correct esoteric-language code. As shown in Figure 6, there is no clear downward trend in EsoEval accuracy as language obscurity increases, indicating that factors other than raw online prevalence (for example, syntactic similarity to familiar languages or the inclusion of documentation in the prompt) are far more predictive of a model's success.

The lack of a strong correlation between obscurity and model performance implies the difficulty of code generation in these languages is not primarily driven by their rarity or the amount of publicly available information. This finding is somewhat surprising, as one might expect that languages with fewer online resources—such as documentation, tutorials, and example programs—would pose greater challenges for large language models trained on publicly available code.

Unlike widely used languages such as Python, which appear extensively in open-source code repositories, educational materials, and programming discussions, esolangs are mostly confined to niche communities. The lack of formal, structured learning resources in training thus limits the ability of models to generalize from available examples.

## 5 FUTURE WORK

We evaluated four esoteric programming languages—Minipy, Pyth, 0815, and Rhokell—but there are hundreds more. Future work should extend our framework to include additional esoteric programming

languages to verify whether our in-context iterative improvement generalizes across the a broader spectrum of esolangs.

Likewise, our model comparison was limited to two parameter scales of the same family—GPT-4o and GPT-4o-mini—and a handful of open-source counterparts. A more thorough investigation should chart performance across a wider range of model sizes, architectures, and pretraining corpora to uncover any scaling laws specific to esoteric code generation. For example, do larger models show proportionally greater gains on highly unconventional languages, or is there a point of diminishing returns? How do model families with different pretraining objectives (e.g., code-focused versus general-purpose) compare?

While we demonstrated that a handful of well-chosen examples can saturate the model's context-driven gains, we did not optimize which examples to include or how to order them. Future work should explore adaptive retrieval mechanisms that dynamically select the most relevant examples based on the structure and complexity of the target problem Perhaps starting with very simple, canonical exercises and gradually increasing difficulty—may further enhance the model's ability to generalize to specialized domains.

We want to highlight the benefit of extending these optimizations beyond code generation to other low-resource domains, such as symbolic reasoning, formal verification, and theorem proving, which could offer broader insights into the principles governing in-context learning across different tasks.

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

# A ADDITIONAL LANGUAGE OBSCURITY INFORMATION AND ANALYSIS

## A.1 MODEL FAMILIARITY DISCUSSION

Here we provide more detail on the responses summarized in Figure 1. ChatGPT-4o and its mini variant both correctly identified Pyth as a Python-inspired golfing language but showed no genuine familiarity with Rhokell or 0815 and only minimal awareness of Minipy. Deepseek V3 properly classified Pyth and Rhokell while offering only generic or erroneous descriptions for 0815 and Minipy. LLAMA-3.3-70B accurately labeled Pyth and Minipy but failed to provide substantive information on Rhokell or 0815. When presented with five representative code snippets for each language, all models misclassified every example. We observed some common mistakes were interpreting Minipy as buggy Python, labeling Rhokell as Unlambda or vague "functional logic," and giving only superficial labels for Pyth and 0815—thereby demonstrating a marked inability to recognize these esoteric languages from source code alone.

## A.2 ADDITIONAL FIGURES ON OBSCURITY REGRESSIONS

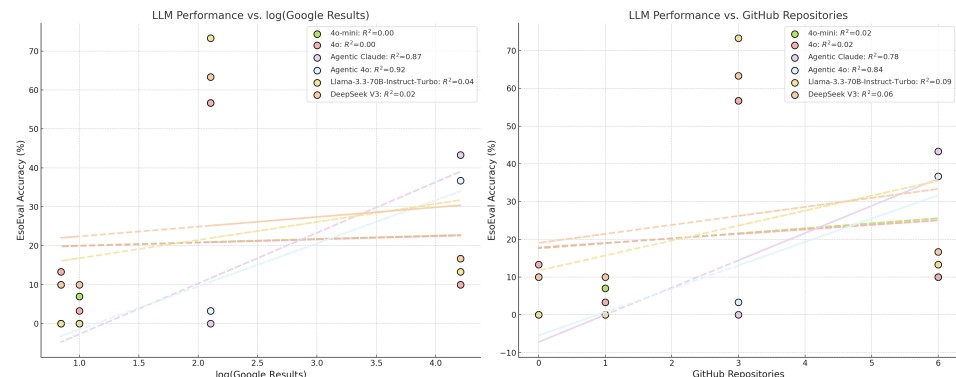

# B IN CONTEXT EXAMPLES

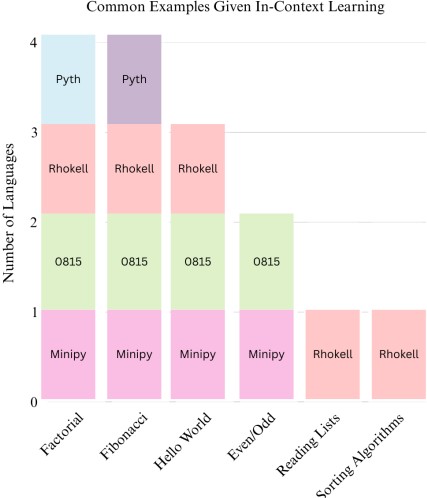

Figure 7: Distribution of common examples provided during in-context learning across the different esoteric programming languages.

For the Esolang 0815 (13 examples) we included "Hello, World!," parity testing, factorial, Fibonacci, sum of squares, "99 Bottles," primes, Hailstone, a randomizer, and truth machines (numeric/ASCII)—four overlap EsoEval, none overlap HumanEval. Pyth (7 examples) featured three factorial variants, subsets memoization, reduce, Fibonacci, and Collatz—two overlap EsoEval. Rhokell (11 examples) covered factorial, Fibonacci, primes, Kolakoski, quicksort, list ops, Peano and binary arithmetic, and a quine—three overlap EsoEval. The examples were gathered from available open-source software examples.

0815

A total of 13 sample programs were provided alongside the 0815 documentation to reinforce model understanding. These included basic outputs like "Hello World!" and Cat, computational tasks such as odd/even checks, binary representation, factorial sequences, arithmetic mean, Fibonacci, and summing squares. More complex problems included "99 Bottles of Beer", prime numbers, the Hailstone sequence, a simple randomizer, and truth machines (numeric and ASCII). There is no overlap between the examples provided and HumanEvaltest set but there is minor overlap between the examples given and those in EsoEval. There is overlap between the in-context examples and EsoEval for the following 4 examples: printing "Hello World!", even/odd number function, factorial, and Fibonacci.

PYTH

For Pyth, a total of 7 examples were provided, with a strong focus on factorial computation. This included three factorial-related sub-examples: Factorial 3.1.1, Factorial 3.1.3 (The Iterative Factorial), and the Recursive Factorial. Additionally, there were examples showcasing memoization (subsets function), functional programming with reduce, Fibonacci sequence generation, and solving the Collatz sequence. There is no overlap between the examples provided and HumanEvaltest set but there is minor overlap between the examples given and those in EsoEval. There is overlap between the in-context examples and EsoEval for the following 2 examples: factorial and fibonacci.

RHOKELL

For Rhokell, a total of 11 examples were provided, covering a range of algorithmic and computational topics. Several examples focus on mathematical sequences, such as computing factorials, Fibonacci numbers, primes, and the Kolakoski sequence. Sorting and list manipulation are also demonstrated, with a quicksort implementation and a general lists example. Additional examples explore syntax and functional programming concepts, including Peano arithmetic, binary arithmetic, and a quine program. There is no overlap between the examples provided and HumanEvaltest set but there is minor overlap between the examples given and those in EsoEval. There is overlap between the in-context examples and EsoEval for the following 3 examples: printing "Hello World!", factorial, and Fibonacci.

MINIPY

For Minipy, no examples were provided for EsoEval. However, among the code generated by Deepseek V3 for EsoEval—which compiled correctly only in the Minipy interpreter and not in the standard Python interpreter—the resulting examples were collected and subsequently used for testing on HumanEval. There is no overlap between the examples provided and those in HumanEval.

## C  AGENTIC AI FRAMEWORK EVALUATION

For the second part of our evaluation, we turned to agentic AI frameworks. Specifically, we evaluated tools such as Windsurf and Cursor by prompting their respective agents to write the code for both the HumanEval and EsoEval datasets.

### C.1  LANGUAGE PARSING

One additional challenge arose from the need to adapt our testing harness to the specifications of each language and the structure of the HumanEval benchmark. Because HumanEval's reference

implementations use Python-style assert statements, any candidate solution needed a a Python-callable function. In practice, many esoteric-language programs required input-output wrappers to conform to the HumanEval harness, and some languages lacked any built-in notion of user-defined functions. We translated between string, list, or integer representations —to ensure that each candidate program could be tested uniformly by the test runner. At the same time, we strove to respect each language's native syntax and execution model, providing only the smallest necessary adaptation rather than rewriting the core logic. As a result, every testing harness is slightly different; in the remainder of this section, we describe those per-language adjustments in detail

### C.1.1  MINIPY

MiniPy was the most straightforward language to work with due to its similarity with Python as a coding language. For this language, the outputed code was directly executable using a Python compiler with a list of shorthands appended to the beginning of each program.

### C.1.2  PYTH

**Architecture Overview**   We developed a systematic approach to testing Pyth code using Python's testing infrastructure, focusing on three key components. The first component was our Code Translation Layer, which implemented `get_pyth_translation` to capture Python translations from the Pyth interpreter's stderr output. This was important since our testing dataset contained our tests using Python assert statements. Therefore, parsing the translation in the stderr output was the simplest solution.

The second component, our Test Execution Environment, centered around the `test_pyth_function` which dynamically executed Pyth code with arbitrary inputs. Since the output of the Pyth program only existed withint the context of the interpreter, we set up an environment to manage variables through a global `environment` dictionary and handled return value propagation via `environment['K']`, ensuring consistent state management between Pyth and Python contexts.

The execution flow is shown in the following workflow:

```
def workflow(pyth_code, input_value):
    translation = get_pyth_translation(pyth_code)
    python_func = create_python_function(translation)
    result = test_pyth_function(python_func, input_value)
    return result
```

### C.1.3  0815

**Architecture Overview**   For the 0815 esoteric language implementation, modified the testing framework to address the differences with working with a register-based hexadecimal language. The first component was our Register Management System, which handled the language's three 64-bit registers: X (write-only), Y (helper), and Z (read-only). This involved state tracking and hexadecimal conversions for all numeric operations.

For Test Case Integration, we implemented a system that bridged between decimal test inputs and 0815's hexadecimal requirements. This included automatic conversion of test inputs to hexadecimal format and proper interpretation of hexadecimal outputs back to decimal for test validation. This was especially important when figuring out representations for lists and other unique data structures. We also refactored the assert statements within the HumanEval test cases to generate text files with the test cases written out instead. They were then parsed and converted using the process described above to test each program.

### C.1.4  RHOKELL

**Architecture Overview**   For the Rhokell language implementation, we developed a testing framework that integrated with Rust's cargo build system while providing a Python-based test harness. The first component was our Rust Integration Layer, which managed the compilation and execution

of Rhokell code through cargo. This required careful handling of build processes and proper path management to ensure reliable interpreter access.

The second component was our Execution Environment, which utilized a robust subprocess management system to handle both compilation and runtime phases. This dual-phase approach was necessary due to Rhokell's compiled nature, distinguishing it from interpreted languages like Pyth and 0815. The environment tracked compilation success separately from execution results, providing detailed feedback for both phases.

For Test Case Management, we implemented a dataclass-based statistics tracking system that monitored multiple aspects of test execution. This included tracking total problems attempted, successful compilations, passed tests, and aggregate test counts, providing comprehensive metrics for evaluation.

**Key Technical Challenges and Solutions**  The implementation presented several unique technical challenges. The primary challenge was managing the Rust-based interpreter's build process. Unlike the other esolangs we tested, Rhokell was not implemented with Python-based interpreters. We resolved this by implementing a pre-execution build check system that verified the interpreter's availability and triggered compilation when necessary.

This also involved delving into the process management, especially with testing. The solution involved implementing a timeout-aware execution system that properly handled both compilation and runtime errors while maintaining clean state. The test cases were treated similarly to previous esolangs, being written into a text file and then parsed into a form recognizable by the language.

## D    REPRODUCIBILITY AND LLM USAGE

We intend to make the code used to run these experiments available online along with the paper. The use of proprietary LLMs and agentic frameworks may make impede reproducibility if the exact versions used for these tests are no longer available. We have documented testing dates and versions to help assess this. The github obscurity measure should be possible to check at a given date, the search result numbers are likely not able to be reproduced.

We have used LLMs to assist in writing code and editing the paper. Any LLM output used has been reviewed by authors on this paper.

