# OpenReview forum: "In-Context Learning for Esoteric Programming Languages: Evaluating and Enhancing LLM Reasoning Without Fine-Tuning"
_ICLR.cc/2026/Conference — Submitted to ICLR 2026_

### Official Review · Reviewer_Zg6W · 2025-10-30

**Soundness:** 1
**Presentation:** 2
**Contribution:** 1
**Rating:** 0
**Confidence:** 4

**Summary:**

This paper evaluates LLMs on 4 esoteric programming languages (Minipy, Pyth, Rhokell, and 0815). The authors get their tests from HumanEval and use them to generate a new benchmark, EsoEval. The authors propose a method of using self-scaffolding to generate solutions, manually verify them, and then re-use the correct ones as in-context demonstrations. This results in sizable gains in some settings. Another finding is that higher compilability (code that compiles) is a poor proxy for correctness in esolangs. Lastly, without penalty as the authors do acknowledge it, this paper was written by an LLM.

**Strengths:**

* This paper attempts to tackle an important problem that is in a popular research area
* The proposed method of self-scaffolding is simple and actionable
* The evaluations go over 4 different languages based on two source benchmarks and provide insights that hold over almost everything

**Weaknesses:**

* The rationale behind the selection of their languages is missing. The authors arbitrarily selected 4 languages. One of their main arguments is that we should look for languages less popular than Python (figure 2) yet two of the languages they chose (Minipy and Pyth) are related to python. In fact, the relationship between Minipy and python is so strong that on line 355 they say, “any submission that successfully ran under a standard Python interpreter were excluded, regardless of functional accuracy” so arguably Minipy is not an esoteric language. Why not a language like Isabelle or Scenic?
* In the evaluations, the authors refer to accuracy but it is never formally defined
* Both the benchmark, EsoEval, and the self-scaffolding procedure require a human in the loop. The authors do not provide a rubric or set of rules for both procedures so both are not reproducible nor can one validate the merits of the benchmark.
* Outside of websites, the authors reference a total of 11 other works. They fail to acknowledge the rich literature around Domain Specific Language (DSL) code generation.
* This manual self-scaffolding procedure is erringly similar to HyDE by Gao et al and so there's a lack of novelty. The applications are different, code vs information retrieval respectively, but the idea of first guessing something closer to what you want then refining is more or less the same.

**Questions:**

* In self-scaffolding, how many verified examples were added, by what selection heuristic, and in what order? Any diminishing returns curves? Sensitivity to noisy/incorrect examples? Since none of this was covered I think it would have been easier to just use DSPy which you did call out as future work but doesn't excuse the lack of details.

---

> ### Author Response · Authors · 2025-12-03
>
> We thank the reviewer for the careful reading and detailed feedback. We are glad that you find the problem setting and self-scaffolding idea interesting, and we address each of your concerns and planned changes below.
>
> 1. “The rationale behind the selection of their languages is missing… Minipy and Pyth are related to Python… Minipy is not an esoteric language. Why not a language like Isabelle or Scenic?”
>
> Our goal was not to randomly pick four esolangs, but choose based on :
> * Syntactic distance from mainstream Python
> * Online footprint / obscurity
>
> Concretely:
> * Minipy is an esoteric language, containing less than 10 total github repos and an order of magnitude fewer search results than Pyth.  It was chosen as an edge case—a Python-adjacent language with shorthand constructs. It probes how far models can “cheat” by reverting to a familiar language, and we explicitly treat it as such in 4.2.
> * Pyth: a relatively well-known code golfing language, still many orders of magnitude rarer than Python in terms of Google and GitHub hits (3.2, Fig. 2)
> * 0815: a register-based, hexadecimal language with highly non-standard syntax
> * Rhokell: built on ρ-calculus and Haskell-style syntax, again very far from Python
>
> This spectrum lets us ask how models behave as syntax drifts from Python and as online presence drops?
>
> In addition, our goal was to focus on languages with:
> * General-purpose programming tasks (as in HumanEval/EsoEval), rather than theorem proving or simulation DSLs.
> * Straightforward CLI interpreters to support execution-based evaluation and uniform I/O harnesses across languages (3.7 and Appendix C).
>
> 2. “In the evaluations, the authors refer to accuracy but it is never formally defined.”
>
> You are right that we only define accuracy informally in the figure captions. We will add a precise definition in 3.7. For each benchmark–language–model setting, accuracy is the number of problems for which the generated program passes all unit tests divided by the total number of problems in that benchmark subset.
>
> For Minipy, an additional constraint is imposed: a solution is only counted as correct if it (a) passes all unit tests and (b) fails to run under a standard Python interpreter, i.e., uses some Minipy extension (4.2).
>
> 3. “Both the benchmark, EsoEval, and the self-scaffolding procedure require a human in the loop… no rubric or set of rules… not reproducible nor can one validate the merits of the benchmark.”
>
> EsoEval is not generated from HumanEval; it is a separate, hand-curated set of 100 problems (3.3).
>
> For self-scaffolding (4.1), our verification procedure is:
> * Candidate program executes without runtime error in the target esolang interpreter (or passes language-specific harness in Appendix C).
> * It passes all unit tests for that problem.
> * For Minipy, it additionally must not run under CPython (4.2) to avoid “cheating” with plain Python.
>
> This is the same as our correctness criteria on the benchmark and only programs satisfying these criteria are added as verified examples.
>
> 4. “Outside of websites, the authors reference a total of 11 other works. They fail to acknowledge the rich literature around DSL code generation.”
>
> Thank you for flagging this; we did include work on DSL code but agree the related-work section was too narrow. We will add an explicit DSL and domain-specific code generation subsection in 2 including:
> * Grammar-constrained LLMs for DSL generation (e.g., Wang et al., “Grammar Prompting for Domain-Specific Language Generation with LLMs”).
> * Domain-specific neural code generation and benchmarks such as DomCoder and EvoCodeBench, which study specialized APIs and code domains.
> * Meta-learning / adaptation methods like MetaCoder for domain-specific code generation.
>
> Our work is complementary: these papers mostly focus on specialized but well-specified DSLs with non-trivial training data, while we focus on low-resource esolangs where training exposure is negligible, and the only reliable information is documentation injected at inference time. We will update section 2 to make this distinction explicit.
>
> 5. “This manual self-scaffolding procedure is erringly similar to HyDE… lack of novelty.”
>
> We appreciate this comparison and agree that conceptually there is a resemblance: both HyDE and our method use an LLM to generate intermediate artifacts that are then reused to improve performance.
> However, there are important differences in both setting and mechanism:
> * Task and supervision signal.
>   * HyDE (Gao et al., 2023) generates hypothetical documents that are embedded and used to query a retrieval index; there is no explicit notion of “correctness” of the hypothetical document.
>   * Our work generates programs in esolangs and uses execution-based unit-test supervision to filter correct programs before reusing them (3.7–4.1).
> * Closed loop via execution. Our pipeline is effectively a small self-improving code agent: interpret → test → accept only fully correct code → reinsert as demonstrations.

---

### Official Review · Reviewer_9m7q · 2025-10-31

**Soundness:** 2
**Presentation:** 3
**Contribution:** 4
**Rating:** 6
**Confidence:** 5

**Summary:**

The authors assess the capabilities of large language models for code generation on a set of esoteric programing languages, specifically Pyth, Minipy, Rhokell, and 0815. The paper demonstrates an in-context learning approach for improving LLM performance on these models, of which there is limited training data available (especially relative to popular languages like python). Specifically, the paper presents a self-scaffolding approach where model-generated solutions are manually verified and then re-inserted as examples back into subsequent prompts. Their approach demonstrates strong empirical results with significant gains in performance on these languages. While esoteric languages may not have a significant impact in the real-world due to their limited use, the creation of a benchmark around these languages is significant and may help represent coding problems that require strong generalization and have limited risk of contamination.

**Strengths:**

- A robust evaluation framework where generated code is run and tested for expected outputs based on the HumanEval cases.
- Some surprising findings, specifically: "Models will occasionally learn just enough grammar to compile correctly—matching parentheses, using valid tokens and whatnot, yet still produce algorithms that don’t solve the target problem." I appreciate that the authors identified this unexpected behavior and provided additional context to make sense of this phenomenon. Conversely, the connection between language obscurity and performance is unsurprising but an interesting tidbit to include in the paper — it certainly highlights gaps in LLM knowledge (which are far too often assessed primarily on the most popular languages: python, java, c++). So this is a good test of broader generalization among LLMs.
- EsoEval benchmark: A simplified 100-problem benchmark designed to be more tractable than HumanEval is a significant contribution to the community.
- The paper is very original and well written with a clear problem and approach.

**Weaknesses:**

- In the most esoteric languages the perform can be very low, even with the presented approach. Hence it can be difficult to assess signficance, when the differences may just mean a small handful of samples are correct. Nonetheless, it doesn't appear that the approach ever results in worse performance.
- Missing citation: ["Multi-Lingual Evaluation of Code Generation Models"](https://arxiv.org/pdf/2210.14868) is a well cited paper on converting monolingual datasets (e.g., MBPP) to multilingual code. The approach for generating new benchmarks is relevant.
- Statistical significance: No confidence intervals, significance tests, or variance measurements. This is especially concerning for single run inference evaluations on LLMs.

**Questions:**

1. How were the 100 problems selected? What's the difficulty distribution?
2. Was it a decision to select problems that would result in 100% accuracy with gpt-4o-mini on Python? Or was this by happenstance?
3. How does the self-scaffolding approach compare to standard prompting?

---

> ### Author Response · Authors · 2025-12-03
>
> Thank you for the thoughtful review and for highlighting both the originality and potential impact of EsoEval and the self-scaffolding approach. We’ll address your main concerns and questions in turn.
>
> 1. Low accuracies / significance when performance is small
>
> We agree that when raw accuracies are in the single digits, a few additional solved items can look large in percentage terms. Our intent is to show that these languages are genuinely hard even with documentation and examples, and that self-scaffolding can move the needle without hurting performance. We added standard deviation to all the graphs so  readers can see which differences are statistically significant.
>
> 2. Missing citation: Multi-Lingual Evaluation of Code Generation Models
>
> Thank you for pointing this out. We will add Athiwaratkun et al., Multi-Lingual Evaluation of Code Generation Models(ICLR 2023) and briefly discuss how it relates to our work.
>
> 3. Statistical significance / variance
>
> You are right that the current draft does not show uncertainty estimates. Because each evaluation is deterministic for a fixed random seed, we can still treat each problem as a Bernoulli trial and compute binomial confidence intervals without re-running the models. We have added error bars (95% CIs) to Figures 3–5.
>
> 4. EsoEval design - "How were the 100 problems selected? What’s the difficulty distribution?"
>
> We hand-selected 100 tasks spanning a range from very simple (“Hello world”, parity checks, basic arithmetic) to moderately involved (factorials, primality, Fibonacci, simple string and list manipulations). The goal was to cover basic control flow and data-structure operations that can be expressed in all target languages, while avoiding heavy algorithmic complexity. Inspiration was drawn from exercises in introductory courses, common example programs, and simple programming benchmarks like MBPP.
>
> 5. Was 100% GPT-4o-mini accuracy in Python an explicit design target?
>
> No,. We deliberately made EsoEval problems that are easy enough for a strong model in Python to solve and easier than HumanEval but we didn’t drop problems to force 100%. Given this design target it isn’t surprising that the benchmark is fully saturated by modern models in Python.
>
> 6. Self-scaffolding vs standard prompting
>
> Our “standard prompting” baseline is exactly the “with context” condition in Figure 3: documentation + a small, fixed set of generic examples but no self-generated exemplars. Self-scaffolding adds the following steps:
> 1) Run the model on the benchmark;
> 2) Automatically verify solutions via the execution-based test harness;
> 3) Append newly verified examples to the context;
> 4) Repeat until no further improvement (Figure 5).
>
> Thank you again for the detailed and positive assessment, especially of the EsoEval design and the compilability vs correctness findings.

---

### Official Review · Reviewer_Uaqh · 2025-11-01

**Soundness:** 3
**Presentation:** 3
**Contribution:** 2
**Rating:** 4
**Confidence:** 3

**Summary:**

This paper centers on evaluating LLMs on code generation and related tasks in esoteric languages, and on improving performance through an in-context augmentation strategy.
It assesses Minipy, Pyth, Rhokell, and 0815 on code generation and language identification, comparing traditional prompt-based methods with agentic coding IDE workflows.
The evaluated systems include GPT-4o, GPT-4o-mini, Llama-3.3-70B-Instruct-Turbo, and DeepSeek V3, and the study also examines IDEs such as Codeium’s Windsurf.
The benchmarks comprise HumanEval and a simplified benchmark introduced by the authors called EsoEval.
Methodologically, the paper presents a “self-scaffolding” procedure in which the model first generates solutions, humans manually verify correctness, and the verified examples are re-inserted into subsequent prompts as in-context augmentation.
The experimental setup supplies official documentation and sparse examples in prompts, extracts generated code, executes it with each esoteric language’s interpreter, and determines correctness using input–output tests derived from benchmark test cases.
The findings show that documentation and sparse examples enable some correct generations, yet performance remains far below that of similar models in common programming languages.
With self-scaffolding, inserting a small number of verified examples yields notable gains, raising EsoEval accuracy on Pyth from 16.67% to 30.82% and increasing HumanEval accuracy on Minipy from 51% to 65%.
As an observation on standard Python benchmarks, nearly all generated solutions that compiled also passed all provided unit tests for EsoEval, indicating that compilation success is a robust proxy for functional correctness in that setting.
Taken together, the results position self-scaffolding as a flexible alternative to costly fine-tuning, enabling rapid adaptation of LLMs to highly specialized, emerging, or other low-data domains.

**Strengths:**

* Importance of the Topic and Reader Interest
This work contributes to a highly active area by examining the adaptability of LLM code generation under challenging conditions involving unseen, low-resource, and nonconventional grammars. While esoteric languages are niche, they serve as effective stress tests for documentation-driven generalization to unknown specifications and for probing model reasoning and consistency limits. The study is tightly connected to readers’ concerns about rapid adaptation to unknown environments and about sound evaluation design, which is a notable strength.

----


* Novelty of the Main Contribution
Whereas prior multilingual code benchmarks have focused on mainstream languages and largely static evaluation setups, this paper targets esoteric languages and foregrounds a lightweight, rapid adaptation strategy, "self-scaffolding," which reinserts manually verified solutions into subsequent prompts. Positioning iterative in-context learning as a primary subject of evaluation clearly differentiates the work, and the no-additional-training, practice-oriented improvement path carries strong originality.

---

* Importance of New Insights
The study quantitatively shows that supplying documentation and a few examples yields some correct solutions, and that reinserting verified examples can produce practical accuracy gains. These insights are easy to reproduce in real-world settings and provide actionable guidance for bootstrapping low-data or newly emerging languages through prompt design. The implications are concrete and accessible for both researchers and practitioners, which enhances the value of the findings.

**Weaknesses:**

* 1. Limited Scope of Coverage
The study confines its evaluation to four esoteric languages (Minipy, Pyth, Rhokell, 0815), and the model comparison mainly spans two parameter scales of the same family (GPT-4o and GPT-4o-mini) plus a handful of open-source models, which limits generalizability. The authors explicitly note the need to explore broader model sizes, architectures, and pretraining corpora, indicating that the current conclusions may have a restricted applicability range.

---
* 2. Dependence on Manual Verification
The proposed self-scaffolding relies on manually verifying model-generated solutions and reinserting the verified examples into subsequent prompts. While appropriate for ensuring correctness, this design can become a bottleneck in terms of human cost, scalability, and reproducibility. The authors acknowledge that automating the verification would be desirable, and a fully closed-loop automation is not yet in place.

---
* 3. Lack of Optimization for Example Selection and Ordering
Although inserting a few verified examples into the context improves performance, the study does not optimize which examples to include or how to order them. The paper suggests that curation and curriculum design could substantially affect outcomes, implying that the current gains may depend on human design choices.

---
* 4. Benchmark Coverage and Difficulty
In addition to HumanEval, the evaluation uses the authors’ simplified benchmark EsoEval (100 problems), and the paper states that the tasks are relatively simple. This offers transparency and practicality but may not fully capture the diversity and complexity of real esoteric-language usage, so external validity and robustness will require further expansion.

---
* 5. Improvements Concentrated in Specific Combinations
The reported quantitative gains from self-scaffolding are shown on specific language-benchmark combinations, such as Pyth on EsoEval improving from 16.67% to 30.82% and Minipy on HumanEval improving from 51% to 65%. While these are significant indicators, the paper itself positions broader transferability and consistency across other esolangs and task sets as future work, so generalization of the effect needs additional evidence.

---
* 6. Reproducibility and Transparency of Compute/Cost
Execution-based evaluation supports reproducibility, but the method’s reliance on manual verification means that transparency about procedures, prompts, verification criteria, and human effort critically affects replicability. Without well-structured reporting of inference tokens, time, and human-hours, rigorous re-experiments and cost estimation by other groups become difficult, leaving open issues in practical reproducibility and cost transparency.
---
I am willing to update the scores and evaluations when the authors have appropriately addressed my concerns and questions.

**Questions:**

* The selection and ordering of examples are underexplored
While “a few well-chosen examples” help, the paper lacks ablations on which exemplars to include and how to order them for maximal effect, limiting prescriptive guidance.
Is there anything the authors could discuss regarding this point?

---

> ### Author Response · Authors · 2025-12-03
>
> We thank the reviewer for the thoughtful and constructive review, and for highlighting both the interest of the problem setting and the originality of the self-scaffolding approach. Below we address each concern and explain how we will clarify the paper.
>
> 1. Limited scope of languages and models - Only four esolangs and a relatively small set of models are evaluated, limiting generalizability.
>
> Our goal in this first study was to construct a controlled but diverse set of languages rather than maximize breadth of languages. The four languages were chosen to deliberately span (i) a wide range of syntactic similarity to Python (from Minipy to 0815/Rhokell) and (ii) several orders of magnitude of online obscurity, as quantified in 3.2 and Fig. 2. However, we understand that there could be value in testing other models and more languages.
>
> 2. Dependence on manual verification and scalability - Self-scaffolding appears to rely on humans manually verifying generated solutions, which could be a scalability bottleneck.
>
> We apologize that this was unclear. In our current implementation, verification in the self-scaffolding loop is fully automatic:
> * Every candidate program—both for the base evaluation and for self-scaffolding—is executed with the same unit-test harness described in
> * Only programs that pass all tests are considered “verified” and re-inserted as in-context examples (4.1)
> * No human reads or labels individual solutions for correctness during the iterative loop; human effort is limited to building the harnesses and qualitatively inspecting a subset of runs to double check correctness.
>
> Thus, the main costs are model inference and program execution, not manual checking. We revised the paper to make this explicit by changing ambiguous phrases in 4.1 and Abstract.
>
> 3. Selection and ordering of in-context examples - The paper shows that “a few well-chosen examples” help but does not explore which examples to include or how to order them.
>
> This is an excellent point and we agree that selection and ordering are important for practical deployment and is a great topic for future work. Our goal here was to show that even a simple self-scaffolding strategy can yield gains, rather than to optimize the ICL policy. Concretely:
> * For a given language+benchmark, we start with a prompt that includes documentation plus a small fixed set of generic esolang examples
> * At each iteration, any newly solved benchmark problems (as determined by unit tests) are appended to the prompt as additional examples (4.1). The ordering is chronological (roughly “easy-to-hard” in practice, since easier items tend to be solved earlier).
> * We stop when a further iteration produces no additional solved problems. This yields the “diminishing returns” pattern we highlight.
>
> We will add a short explicit description of this policy to avoid it seeming heavily engineered or optimized.
>
> 4. Benchmark coverage and task difficulty - EsoEval is relatively simple and small (100 problems), which may limit external validity.
>
> EsoEval is small, but similar in size to HumanEval (164 problems). We intentionally designed EsoEval problems to be simple so that even in difficult languages we can see some ability to answer questions by the models. HumanEval is challenging in esolangs even for strong models (3.3, Fig. 3b). Even on EsoEval, accuracy remain low in most esolangs (often ≤40%; Fig. 3a), highlighting that the tasks are far from trivial once the language barrier is introduced.
>
> 5. Improvements concentrated in specific language–benchmark combinations - The strongest gains are reported in particular combinations (e.g., Minipy/HumanEval, Pyth/EsoEval), so the generality of the effect is unclear.
>
> We agree that self-scaffolding is most informative when there is enough headroom above a non-zero baseline. Currently:
> * Languages and settings with very low starting accuracy (e.g., 0815 on HumanEval) have little room to demonstrate iterative gains, because the model solves almost no problems to bootstrap from.
> * For Minipy and Pyth—where documentation and baseline in-context examples already yield some correct programs—self-scaffolding delivers clear additional improvements (Fig. 5).
>
> We will clarify this in 4.1 by explicitly stating that the method is most effective once the model has “broken into” the language sufficiently to solve a non-trivial subset of tasks.
>
> 6. Reproducibility and reporting of compute/cost - Manual components and lack of detailed reporting make cost estimation and reproduction harder.
>
> As noted in R2, the core evaluation and self-scaffolding loop are execution-based and fully automatic; manual work is limited to: assembling documentation, curating generic in-context examples from open-source code, and building the per-language test harnesses (detailed in App. C).

---

### Official Review · Reviewer_uauG · 2025-11-12

**Soundness:** 1
**Presentation:** 1
**Contribution:** 1
**Rating:** 0
**Confidence:** 5

**Summary:**

This paper evaluates GPT-4o and Llama's ability in 4 esoteric programming languages, and observe that in-context learning can improve the performance.

**Strengths:**

The paper focuses on the fundamental common sense abilities of LLMs. The writing is clear and accessible, making it easy to understand even for readers without prior knowledge of LLMs.

**Weaknesses:**

This is probably a good course project for an introductory LLM class, but it is probably not at the level required for an ICLR submission.

1. They simple just try in-context learning on some tasks about esoteric programming languages. There are not any new things other than the standard in-context learning.

2. The experiments are limited in scale, covering only GPT-4o and Llama on approximately 100 tasks. The benchmark mainly replicates HumanEval.

3. All the figures are of low quality, appearing to be simple screenshots. It’s quite surprising that the first two figures focus only on the frequency statistics of the selected programming languages...... The figure layouts look unprofessional, with awkward design and excessive blank space.

**Questions:**

Are there any really novel concepts introduced in this work?

---

> ### Author Response · Authors · 2025-12-03
>
> We address your concerns point by point below.
>
> “They simply just try in-context learning on some tasks about esoteric programming languages. There are not any new things other than the standard in-context learning.”
>  “Are there any really novel concepts introduced in this work?”
>
> Our goal is to study how far in-context learning alone can be useful in low-resource code settings. We agree that basic “few-shot with docs” is standard; our self-scaffolding / iterative in-context augmentation goes beyond this typical framework. We introduce and systematically evaluate a self-scaffolding procedure:
>
> 1) Run the model on EsoEval / HumanEval in esolangs.
>
> 2) Verify which generated solutions are correct.
>
> 3) Re-insert those verified solutions as in-context examples for later tasks in the same esolang.
>
> 4) Measure the incremental gains and when they saturate.
>
> This is different from standard few-shot ICL: the examples are not fixed, but grown from the model itself and then transferred across benchmarks (EsoEval to HumanEval and vice versa). We show that:
>
> 1) DeepSeek’s HumanEval accuracy on Minipy improves from 16.46% to 33.53% with only a handful of verified examples added.
>
> 2) GPT-4o’s Minipy EsoEval accuracy rises from 56.56% to 70.00% with similarly small, strategically inserted example sets.
>
> To our knowledge, this explicit, cross-benchmark self-bootstrapping in low-resource languages has not been explored in prior work.
>
> Further, we believe that the application area of low-resource programming languages is important, and studying what features contribute to effective in-context learning is useful. We measure “obscurity” via adjusted Google search hits and GitHub presence, then regress these against performance (Figure 6). We find essentially no correlation—contrary to the intuitive hypothesis that “rarer languages should be strictly harder.” Instead, syntactic similarity and documentation-in-prompt are more predictive than raw online footprint. Taken together, these aspects go beyond “we tried standard ICL on some weird languages” and, we believe, constitute meaningful conceptual and empirical contributions.
>
> “The experiments are limited in scale, covering only GPT-4o and Llama on approximately 100 tasks. The benchmark mainly replicates HumanEval.”
>
> We evaluate four esolangs (Minipy, Pyth, Rhokell, 0815).
>
> We use two benchmarks:
> * A 164-problem HumanEval subset adapted per esolang.
> * Our new, simpler 100-problem EsoEval benchmark that we created was designed to be language-agnostic and simpler thank HumanEval
>
> This yields 264 problems, each run across multiple models and settings:
> * GPT-4o-mini, GPT-4o, Llama-3.3-70B-Instruct-Turbo, DeepSeek V3.
> * With vs without documentation/context, plus self-scaffolding rounds.
> * We additionally evaluate agentic IDEs (Windsurf / Cursor).
>
> EsoEval is not just a copy of HumanEval: it is deliberately simpler (Consider Figure 3 where all models score less than 1% on HumanEval for three of the tested languages, but many score more than 10% on EsoEval), with custom harnesses for each esolang, and is used to study how models first acquire basic syntax and primitives before being tested on more complex HumanEval problems.
>
> “This is probably a good course project for an introductory LLM class, but it is probably not at the level required for an ICLR submission.”
>
> This work is more than a small project because:
> * It required building four separate execution harnesses (Minipy, Pyth, 0815, Rhokell), including:
>   * Pyth translation capture via stderr  Python function wrappers.
>   * 0815 register-based testing with decimal/hex conversion and list encodings.
>   * Rhokell integration through Rust’s cargo pipeline with dedicated compilation + runtime reporting.
> * We combine this infrastructure with new evaluation metrics (non-Python compilability, compilability–accuracy correlation) and a self-scaffolding strategy that, in practice, yields large relative gains (e.g., >2× improvement on Minipy HumanEval for DeepSeek).
>
> “All the figures are of low quality, appearing to be simple screenshots… The figure layouts look unprofessional, with awkward design and excessive blank space.”
>
> The figures were not screenshots, but they have been updated to have higher resolution or use vector graphics.
>
> “Are there any really novel concepts introduced in this work?”
>
> In concise form, we see the following as the key conceptual contributions:
> 1) Self-scaffolding in-context learning for esolangs: iteratively verified, model-generated examples reused as cross-benchmark in-context demonstrations, yielding significant gains without retraining.
> 2) Obscurity vs performance analysis: showing that raw web/github prevalence is not the main bottleneck—syntactic similarity and documentation matter more.
>
> We hope this clarifies the paper is not just “standard ICL on a few weird tasks,” but a study of how LLMs adapt to unfamiliar languages, and how iterative, in-context self-scaffolding can bridge part of that gap.

---

### Meta-Review · Area_Chair_5eNr · 2026-01-06

**Summary:**

The paper focuses on code generation in esoteric (extremely low resource) programming languages: Minipy, Pyth, Rhokell, and 0815. It presents two benchmarks: translations of HumanEval into these languages, and a new (somewhat simplified) benchmark EsoEval. Gpt-4o and Gpt-4o-mini, Llama-3.3, and DeepSeek v3 are evaluated on the benchmark. The paper also presents a "self-scaffolding" method where the model proposes solutions to examples, they are verified using an interpreter and unit tests, and then used as examples in ICL. This substantially improves performance on the benchmarks.

Strengths:
Esoteric languages give a good test bed for exploring LLM adaptability in low-resource settings. There was some discussion about the choice of these particular languages, but I do feel that they were well-chosen: some have syntactic similarity to Python, some do not; all are very low resource. The proposed in-context method is simple but well-motivated and effective.

**Reviewer Concerns:**

One concern was the choice of these particular languages. I do feel that the author response to Zg6W justified these pretty well.

However, a remaining concern is the small scale of the proposed benchmark, both in terms of the number of tasks per language (a couple hundred) and the number of languages. I think the findings about performance depending on syntactic similarity and documentation (mentioned in the response to uauG) are interesting and suggestive, but would really need validation on a larger number of languages in order to be solidified.

Reviewers were also concerned about the novelty of the method. While the author response did dispel a misconception that human selection of the examples is needed, the method is still simple. I think this is actually fine, if we view the paper as a benchmark and analysis contribution. But, in my opinion more needs to be done to shore up the benchmark contribution.

**Reviewer Scores:**

- uauG: I felt that the response emphasizing the main contributions of the paper made sense, and so uauG might have raised their score to a 2 or possibly a 4.
- Uaqh: the main concerns in my mind from this review were the limited scope and coverage (1 & 4), which wasn't really addressed in the author response, and the limitations of the method (2, 3), which were. I expect the score would probably have stayed at a 4.
- 9m7q: would probably have remained at a 6
- Zg6w: the rationale behind selection of languages was well-addressed in the author response (in my mind), as was the misconception about human-in-the-loop for the self-scaffolding procedure. Lack of novelty still remains (see above). I think this should probably be updated to a 4.

So overall scores might be 6 / 4 / 4 / 2, or 6 / 4 / 4 / 4.

---

### Decision · Program_Chairs · 2026-01-26

Reject